# Conferring the Metabolic Self-Sufficiency of the CAM Plasmid of *Pseudomonas putida* ATCC 17453: The Key Role of Putidaredoxin Reductase

**DOI:** 10.3390/microorganisms7100395

**Published:** 2019-09-26

**Authors:** Andrew Willetts

**Affiliations:** 1Curnow Consultancies, Helston TR13 9PQ, UK; andrewj.willetts@btconnect.com; Tel.: +44-7966-9684-87; 2College of Life and Environmental Sciences, University of Exeter, Exeter EX4 4JG, UK

**Keywords:** cytochrome P450 monooxygenase, putidaredoxin reductase, diketocamphane monooxygenase, flavin reductase, *Pseudomonas putida* ATCC 17453, (*rac*)-camphor, CAM plasmid

## Abstract

The relative importance of camphor (CAM) plasmid-coded putidaredoxin reductase (PdR) and the chromosome-coded flavin reductases Frp1, Frp2 and Fred for supplying reduced FMN (FNR) to the enantiocomplementary 2,5- and 3,6-diketocamphane monooxygenases (DKCMOs) that are essential for the growth of *Pseudomonas putida* ATCC 17453 on (*rac*)-camphor was examined. By undertaking studies in the time window prior to the induction of Fred, and selectively inhibiting Frp1 and 2 with Zn^2+^, it was confirmed that PdR could serve as the sole active supplier of FNR to the DKCMOs. This establishes for the first time that the CAM plasmid can function as an autonomous extrachromosomal genetic element able to express all the enzymes and redox factors necessary to ensure entry of the C10 bicyclic terpene into the central pathways of metabolism via isobutyryl-CoA.

## 1. Introduction

It is now well established that bacterial genes specifying catabolic functions against various natural and xenobiotic organic compounds are located on transmissible plasmids (Table 1, [1,2,3,4,5,6,7,8,9,10,11,12]), enabling these extrachromosomal genetic elements to promote both expanded metabolic capabilities of individual species and horizontal genetic transfer of the encoded metabolic competences.

These concepts originated from pioneering biochemical and genetic studies conducted in the late 1960s [13] on the large (533 kbp) transmissible camphor (CAM) plasmid, which was shown to be essential for the degradation of (+)-camphor to isobutyryl-CoA in *Pseudomonas putida* ATCC 17453. Isobutyryl-CoA provides an entry point into the central pathways of metabolism because it is an intermediate in the chromosome-coded pathways for the catabolism of pantothenate and valine in pseudomonads [14]. While subsequent studies of this bacterium confirmed the location of a number of the genes coding for the requisite enzymes on the large transmissible CAM plasmid [1,15], it is only relatively recently that a more extensive reinvestigation of camphor-grown *P. putida* ATCC 17453 by Iwaki et al. [16] has identified the corresponding plasmid-coded loci for almost all of the enzymes and redox intermediates necessary to metabolise both enantiomers of the C10 bicyclic terpenoid to the level of isobutyryl-CoA (Figure 1). However, the two enantiocomplementary diketocamphane monooxygenases (DKCMOs, Steps C and D) represent notable exceptions. Being flavin-dependent two-component monooxygenases (fd-TCMOs, [17]), these isoenzymes are dependent on reduced FMN (FNR) supplied by a separate flavin reductase (FR) activity. Iwaki et al.’s comprehensive study was notable for assigning this function to Fred, a chromosome-coded 36 kDa homodimeric flavin reductase FR isolated from cells during the late log to early stationary phase of growth on (+)-camphor [16]. This was a striking outcome because it served to preclude the ability of the CAM plasmid to function as a catabolically autonomous entity able to promote entry of camphor into the central pathways of metabolism via isobutyryl-CoA. Intriguingly, no reference was made to this strategically important role for Fred when Iwaki and co-workers reported subsequently that a gene coding for an FR had been identified adjacent to the 3,6-DKCMO-coding gene on the CAM plasmid [18]. This unsubstantiated claim, which directly contradicted their own earlier comprehensive plasmid sequence data [16], was subsequently withdrawn [19].

An alternative approach to establishing the metabolic status of the CAM plasmid tested the competence of various enzymes purified from cells of *P. putida* ATCC 17453 harvested at different stages of growth on camphor-based minimal medium to act as an FNR-generating activity able to promote lactone formation by highly purified preparations of the enantiocomplementary DKCMOs [21,22]. These extensive studies confirmed that chromosome-coded Fred is an inducible enzyme that can serve such a role for each isoenzyme, but only to any significant extent (>2% total detected FR activity) in cells in the late log and subsequent stationary phases of camphor-dependent growth (Figure 2), which is an induction profile consistent with Fred being an enzyme involved in secondary rather than primary metabolism [23]. Conversely, the ability to support the biooxygenating activity of both DKCMOs during the early log and mid log phases of camphor-dependent growth was found to be confined exclusively to three alternative FNR-generating enzymes. Two of these enzymes, Fpr1 and Fpr2, were characterised as chromosome-coded ferric-flavin reductases constitutively expressed in *P. putida* ATCC 17453, whereas the third highly purified competent FNR-generating enzyme was confirmed to be putidaredoxin reductase (PdR), a camphor-induced plasmid-coded 48.5 kDa protein that also serves an equivalent well-established role [24] as one of the functioning subunits of cytochrome P450 monooxygenase (cytP450MO), the initiating enzyme of the camphor degradation pathway (Figure 1). PdR was found to function as a source of FNR able to support both enantiocomplementary DKCMOs when tested not only as a highly purified protein but also when functioning as an integrated subunit of the 98 kDa multimeric monooxygenase [22]. The significance of this newly established role of PdR is that it redresses a significant deficiency of Iwaki et al.’s previous study [16], and for the first time it completes the metabolic potential of the CAM plasmid to function as an autonomous extrachromosomal genetic element able to ensure entry of the C10 bicyclic terpene into the central pathways of metabolism via isobutyryl-CoA. In order to confirm this status, it is imperative to demonstrate that *P. putida* ATCC 17453 can grow on camphor-based minimal medium in the absence of functioning activities for the chromosome-coded enzymes Fpr1, Frp2 and Fred, which is the purpose of the present paper.

## 2. Materials and Methods

### 2.1. Bacterial Strains, Culture Maintenance and Growth Conditions

*P. putida* ATCC 17453 (NCIMB 10007, C1-B, PpG1) was maintained on a basal salt medium supplemented with 7.5 mM *(rac*)-camphor as the principal carbon source as fully described previously [22]. Diauxic growth was routinely achieved by culture on an equivalent medium supplemented with 2,5 mM sodium succinate, with succinate being the initial substrate to be progressively depleted. The levels of succinate and (*rac*)-camphor were monitored, respectively, by reverse-phase HPLC (Agilent 1200, Agilent Technologies, Santa Clara, CA, USA) on a Primesep 100 column [25] and GC (Shimadzu GC-14A, Shimadzu Europe, Druisberg, Germany) on a 10% Carbowax 20 M column [26].

Diauxic growth in the presence of Zn^2+^ was tested by aseptically adding 30 μM ZnSO_4_ from a 0.2 M stock solution when a culture sample was in the late log phase of growth on succinate (A_650_nm = 0.5, remaining succinate < 0.5 mM) and about to enter diauxie.

### 2.2. Culture Samples and Extract Preparation to Determine the Effects of Zn^2+^ Addition to the Diauxic Growth Medium

Samples (20 mL) were taken at timed 10 min intervals for 150 min after cultures (+/− Zn^2+^ addition) had entered late log phase growth on succinate and about to enter diauxie, placed in ice and assayed as expediently as possible for culture density (A_650_nm) and residual succinate and camphor. The biomass content of each sample was harvested by centrifugation (10,000× *g* for 15 min at 5 °C (MSE Coolspin 2, MSE, Heathfield, UK), washed with an equal volume of cold Tris-HCl buffer (0.1 M, pH 7.0) and then re-centrifuged. The recovered cells were evenly suspended in 7.5 mL of the same buffer and subsequently sonicated (Soniprep 150, MSE, Heathfield, East Sussex, UK) in ice for 3 × 2 min. The resulting homogenates were centrifuged (20,000× *g* for 15 min at 5 °C) to remove the cell debris, and a sample (1.0 mL) of each resultant supernatant retained as a crude cell-free extract for activity testing (2,5- and 3,6-DKCMOs). Each remaining supernatant was then subjected to two successive rounds of ultrafiltration (15,000× *g* for 15 min at 5°C) using Amicon centrifugal filters with molecular weight cut-off values of 100 kDa and 30 kDa. The resultant fractions (>100 kDa) were discarded, and the high molecular weight (100–30 kDa) and low molecular weight (<30 kDa) semi-purified fractions were then selectively assayed for cytP450MO (M_W_ 98 kDa), 3,6-DKCMO (M_W_ 85 kDa), 2,5-DKCMO (M_W_
*camE_25-2_* 65 kDa, M_W_
*camE_25-1_* 60 kDa), Fred (M_W_ 36 kDa) and the combined activity of Frp1 (M_W_ 27 kDa) plus Frp2 (M_W_ 28.5 kDa).

### 2.3. Purification of FMN-Reductase Activities

Samples of highly purified Fred were prepared at 4 °C using an LKB Biologic FPLC system deploying successive anion-exchange (Mono-Q, Pharmacia, Stockholm, Sweden), negative affinity (Reactive Blue-4-agarose, Pharmacia) and gel filtration (HiLoad 16/60 Superose 12, Pharmacia) columns in the three-stage protocol fully described by Willetts and Kelly [21]. A combined Frp1 plus Frp2 preparation was a by-product of the procedure, and this was resolved into separate highly purified Frp1 and Frp2 activities using the purification protocol deploying successive gel filtration (Sephadex G-25, Pharmacia) and affinity chromatography (Affi-Gel Blue, Bio-Rad, Hercules, CA, USA) steps developed and fully described by Halle and Meyer [27].

### 2.4. Purification of PdR

Samples of highly purified PdR were prepared at 4 °C using a BioLogic FPLC system (BioLogic10, Bio-Rad) fitted with a Q-Sepharose anion-exchange column (Sigma-Aldrich, Dorset, UK) and eluted with a 0–0.5 M KCl gradient as fully described previously [26].

### 2.5. Assays of Enzyme Activities in the Crude Cell-Free Extracts and the Semi-Purified High and Low M_W_ Ultrafiltrates

Enzyme assays undertaken as indicated on the crude cell-free extracts and high M_W_ and low M_W_ ultrafiltrates were conducted by appropriate previously reported and well-established methods. CytP450MO was assayed spectrophotometrically at 340 nm by measuring the rate of (*rac*)-camphor-stimulated oxidation of NADH in the presence of an excess of PdR and putidaredoxin [24]. The enantiomeric 2,5- and 3,6-DKCMOs were assayed by measuring the rate of NADH-stimulated lactone formation from the corresponding diketocamphane substrate by GC (Shimadzu GC-14A, Shimadzu Europe) on a 10% Carbowax 20 M column [21,22]. Fred was assayed spectrophotometrically (Hewlett Packard model 8452A, Hewlett Packard, Palo Alto, CA, USA) under anaerobic conditions by measuring the initial rate of NADH oxidation at 340 nm as previously described [22]. The combined Frp1 plus Frp2 activity was assayed colorimetrically at 535 nm by using the chromogen bathophenanthroline disulfonate (BPS) to measure the formation of Fe(ІІ)-bathophenanthroline disulfonate from FeCl_3_ [28].

### 2.6. Kinetic Studies of Highly Purified Frp1, Frp2, Fred and PdR

Assays on highly purified enzyme preparations were conducted spectrophotometrically at 340 nm under anaerobic conditions by measuring the initial rate of enzyme-catalysed reduction of FMN by NADH as described previously [21,22].

### 2.7. Reproducibility

Where indicated, procedures were repeated a minimum of five times with equivalent cultures and enzyme preparations, and the resultant data are presented graphically with corresponding standard deviation error bars.

### 2.8. Chemicals and General Procedures

Unless otherwise stated, all chemicals, enzymes and reagents were purchased from Sigma-Aldrich (St. Louis, MO, USA) or Thermo Fisher Scientific (Loughborough, UK) and were used without further purification.

## 3. Results and Discussion

Trophophasic growth of *P. putida* ATCC 17453 on (*rac*)-camphor necessitates functioning DKCMOs, which themselves are dependent on an enzyme-generated supply of FNR. The principal sources of FNR throughout the early log and mid log phases of growth are the chromosome-coded ferric-flavin reductases Fpr1 and Frp2 and plasmid-coded PdR, which collectively account for 98% of the total FNR-generating activity (Figure 2). Selective inhibition studies that discriminate between these various alternative FNR-generating activities should offer further insight into their relative contribution to camphor-dependent growth. While growth-based studies dependent on the selective inhibition of PdR are an impractical option due to its additional role as a functional subunit of cytP450MO [24], the known susceptibility of microbial ferric-flavin reductases to inhibition by Zn^2+^ [29,30,31,32] offers a more promising option.

### 3.1. Effect of Zn^2+^ on Purified Preparations of the Frp1 and Frp2 of P. putida ATCC 17453

Highly purified samples of 27 kDa Frp1 and 28.5 kDa Frp2 prepared from (*rac*)-camphor-grown cells following a previously developed protocol [21] were used to test the effect of Zn^2+^ on enzyme activity. The NADH-dependent reduction of FMN by both Frp1 and Frp2 was found to be susceptible to the addition of increasing levels of Zn^2+^ to the reaction mixture (Figure 3). Both enzymes exhibited a progressive pattern of inhibition in response to increasing levels of the divalent metal ion. In each case, the recorded decrease in enzyme activity was hyperbolic, plateauing at approximately 90–95% of the corresponding no-addition controls, suggesting that neither enzyme could be completely inhibited by Zn^2+^.

Kinetic analysis of purified preparations of Frp1 and Frp2 using conventional Lineweaver–Burk plots indicated that, in each case, Zn^2+^ influenced both the maximum velocity (*V_max_*) and the Michaelis constant (*K_m_*) values for FMN reduction (Appendix A), which suggests that Zn^2+^ can act as a mixed-type inhibitor of both enzymes [33]. The slopes of the plots were used to calculate inhibitory constant (*K_i_*) values for Zn^2+^ with respect to FMN (Figure 4A,B), which were similar for both Frp1 and Frp2 (11.2 μM and 14.9 μM, respectively), values which indicate that Zn^2+^ is a very effective inhibitor of both tested ferric-flavin reductases of *P. putida* ATCC 17453.

Conversely, equivalent studies confirmed that equivalent concentrations of Zn^2+^ had no significant effect on the activity of highly purified preparations of both PdR and Fred (Appendix A). 

### 3.2. Effect of Zn^2+^ on Culture Density and the Specific Activities of Key Enzymes throughout Growth of P. putida ATCC 17453 on (rac)-Camphor

Extensive prior studies [21,34,35,36] have established that inoculation of *P. putida* ATCC 17453 into a defined medium containing both succinate and (*rac*)-camphor results in diauxic growth of the culture. The initial progressive depletion of succinate to below a key threshold level of 0.5 mM triggers a diauxic interlude. During the following 30–40 min, the remaining succinate is further depleted to below detectible levels, and concomitantly the CAM plasmid-coded camphor degradation pathway enzymes are induced. The nascent enzyme activities then promote early log trophophasic growth of the culture to recommence exclusively at the expense of the residual (*rac*)-camphor, prior to the subsequent induction of chromosome-coded Fred which only commences when the culture progresses from late log into idiophasic growth [22]. By dividing such a culture into two on entry into the diauxic interlude, then selectively adding Zn^2+^ to one of the aliquots and subsequently monitoring both aliquots concurrently throughout the following early log phase of camphor-dependent growth, it should be possible to establish the relative importance of the Zn^2+^-insensitive PdR subunit of cytP450MO and Zn^2+^-sensitive ferric-flavin reductases Frp1 and Frp2 as putative suppliers of FNR to the enantiocomplementary DKCMOs. The activity levels of the DKCMOs were assessed in the crude cell-free extracts which contained all relevant cognate FNR-generating systems, whereas the semi-purified high molecular weight (M_W_ 100–30 kDa) and low molecular weight (M_W_ <30 kDa) ultrafiltrates were used to distinguish between activities attributable to cytP450MO (M_W_ 98 kDa) and the much lower M_W_ ferric-flavin reductases Frp1 (M_W_ 27 kDa) and Frp2 (M_W_ 28.5 kDa). 

Using this split culture technique, it was found that adding Zn^2+^ to only one of the resultant aliquots on entry into diauxie resulted in no significant differences in a number of key monitored parameters of culture growth throughout the subsequent early log phase stage of camphor-dependent growth by *P. putida* ATCC 17453 compared to the directly equivalent no-addition control (Figure 5A). The almost identical progressive decreases in both the residual succinate and (*rac*)-camphor levels in both cultures are a clear reflexion of this, as are the similar progressive increases in relevant A_650_nm readings, which indicate that biomass production was only decreased marginally (<4%) throughout (*rac*)-camphor-dependent early log phase growth in the Zn^2+^-supplemented medium. Comparative studies of the unfractionated crude cell-free extracts prepared from both cultures confirmed that the titres of 2,5-DKCMO exhibit very similar induction profiles as do those of the enantiocomplementary 3,6-DKCMO (Figure 5B). One significant exception exclusive to the Zn^2+^-supplemented culture was that within 20 min of the addition of the metal ion, the relevant growth medium became visibly yellow-green (λ_max_ 390–400 nm), a phenomenon that gradually increased in intensity throughout the monitored growth period. Although the chromophore responsible was not isolated and characterised, it is likely to correspond to pseudobactin, a non-ribosomal peptide siderophore known to be produced by *P. putida* and a number of other fluorescent pseudomonads [37] in response to any growth condition promoting limited Fe(III) availability [38], including decreased ferric reductase activity induced by Zn^2+^ addition to the growth medium [30,39].

The significance of the very similar DKCMO titres in the unfractionated crude cell-free extracts prepared from both cultures throughout early log phase growth on (*rac*)-camphor is that these fd-TCMOs, which are key essential enzymes in the camphor degradation pathway, must be sourcing correspondingly sufficient amounts of preformed FNR to function at comparatively equivalent levels in the presence and absence of the 30 μM Zn^2+^ supplement in the growth medium. The FNR-generating activity during early log and mid log phase growth of *P. putida* ATCC 17453 on (*rac*)-camphor, prior to the induction of chromosome-coded Fred, is shared almost exclusively (98%) between the two chromosome-coded ferric-flavin reductases Frp1 and Frp2 and CAM plasmid-coded PdR, a functional subunit of cytP450MO (Figure 2). Taking the titre of cytP450MO in the semi-purified high M_W_ ultrafiltrates as reflecting that of PdR itself [22], and assaying Frp1 plus Frp2 as a combined ferric-flavin reductase activity in the semi-purified low M_W_ ultrafiltrates, a comparison of the specific activities of these FNR-generating enzymes in samples prepared from both cultures throughout the early log phase of camphor-dependent growth confirmed that the only functioning FNR-generating activity in both cultures was the PdR subunit of cytP450MO (Figure 5C,D). Significantly, the combined ferric-flavin reductase activity was very strongly inhibited (<5% activity) within 30 min of Zn^2+^ addition to the supplemented culture, a time course which precedes the progression from diauxie to exclusively camphor-dependent growth (Figure 5A). It is likely that this striking pattern of inhibition directly reflects the demonstrated influence of Zn^2+^ on Frp1 and Frp2 (Figure 3 and Figure 4A,B). While not investigated in this study, it is possible that Frp1- and Frp2-generated FNR may serve one or more additional roles in camphor-grown *P. putida* ATCC 17453 besides facilitating activity of the DKCMOs. This may help explain the small (<4%) decrease in biomass yield resulting from the 30 μM Zn^2+^ addition to the culture medium (Figure 5A). It may also be relevant that the divalent metal ion has been reported to have a number of other poorly characterised toxic effects on microbial growth [40,41].

There were barely detectible titres of Fred, the only other FR shown to be able to supply FNR to the DKCMOs [22], in all of the tested split culture samples. This corresponds with its known induction profile in camphor-grown *P. putida* ATCC 17453 as an activity that only commences expression in late trophophase when the majority of the C10 bicyclic terpenoid has been consumed and the activity levels of the DKCMOs have fallen by >90% [21]. Because high levels of Fred are characteristic of idiophase, it is possible that rather than acting to supply FNR to the DKCMOs, the enzyme serves a role in the biosynthesis of secondary metabolites such as the polyketides, arylpolyenes, phenazines, acyl-homoserine lactones and rhamnolipids known to be produced in idiophase by a number of other *P. putida* species [42,43]. Such a role for Fred is also consistent with the almost identical induction profile for the enzyme when *P. putida* ATCC 17453 is grown on an equivalent minimal medium containing 10 mM succinate as the sole carbon source (Appendix A).

Collectively, these outcomes indicate that, in the absence of active titres for the chromosome-coded enzymes Fred, Frp1 and Frp2, the FNR necessary for the efficient functioning of the DKCMOs in camphor-grown *P. putida* ATCC 17453 is supplied principally, if not exclusively, by CAM plasmid-coded PdR. This in turn confirms for the first time the ability of the CAM plasmid to function as a catabolically autonomous unit able to ensure entry of the C10 bicyclic terpene, a peripheral organic compound, into the central pathways of metabolism via isobutyryl-CoA. 

## Figures and Tables

**Figure 1 microorganisms-07-00395-f001:**
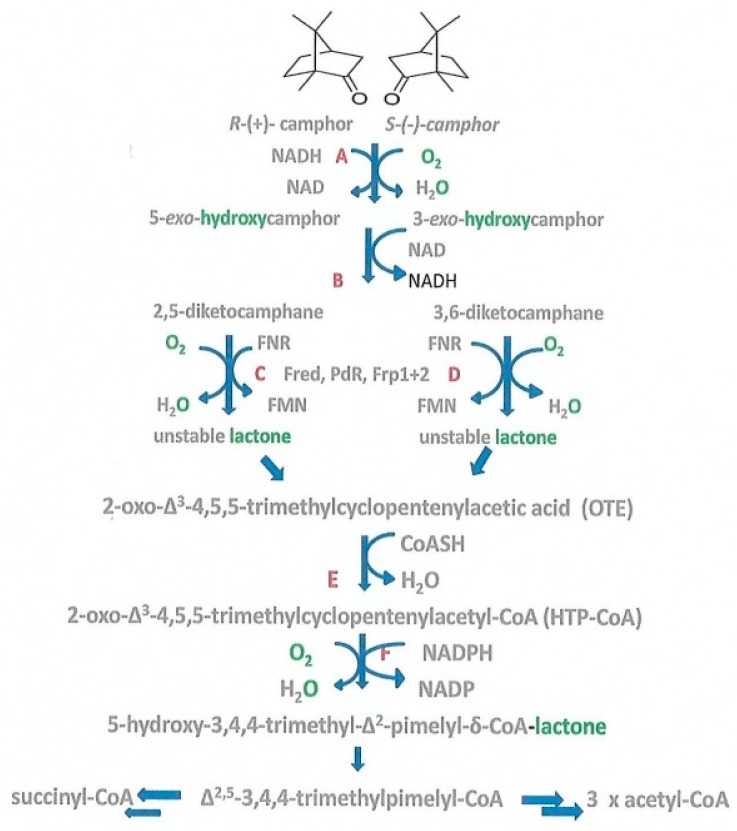
Pathway of (+)- and (−)-camphor degradation in *Pseudomonas putida* ATCC 17453. A, cytochrome P450 monooxygenase (*camCAB*); B, *exo*-hydroxycamphor dehydrogenase (*camD*); C, 2,5-diketocamphane 1,2-monooxygenase (*camE_25-1_* + *camE_25-2_*); D, 3,6-diketocamphane 1,6-monooxygenase (*camE_36_*); E, 2-oxo-Δ^3^-4,5,5-trimethylcyclopentenylacetyl-CoA synthetase (*camF1* + *F2*); F, 2-oxo-Δ^3^-4,5,5-trimethylcyclopentenylacetyl-CoA monooxygenase (*camG*); FNR, reduced flavin mononucleotide; Fred, 36 kDa chromosome-coded flavin reductase; PdR, putidaredoxin reductase subunit of cytochrome P450 monooxygenase (*camA*); Frp1 + 2, chromosome-coded ferric reductases. Diatomic oxygen molecules participating in the four monooxygenase-catalysed steps are shown in green, as in each case are the fates of each component oxygen atom. Adapted from Figure 1 [20].

**Figure 2 microorganisms-07-00395-f002:**
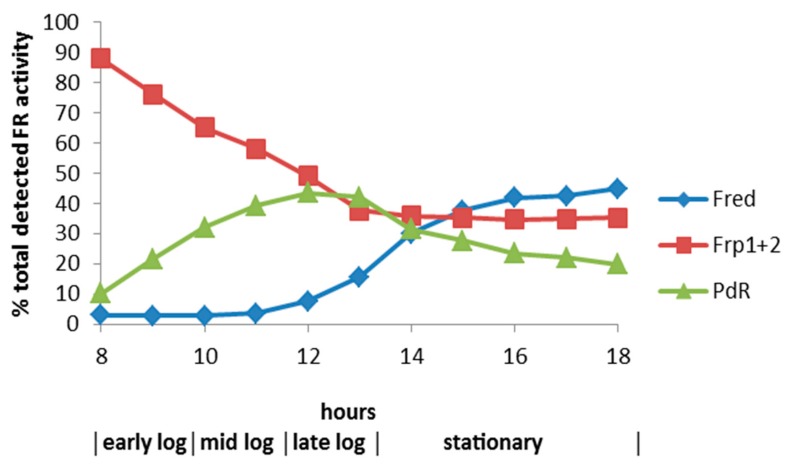
Relative contribution of the different assayed FNR-generating enzymes to the total DKCMO-supporting flavin reductase activity titre throughout the various phases of (*rac*)-camphor-dependent growth of *P. putida* ATCC 17453. Adapted from Figure 7 [20].

**Figure 3 microorganisms-07-00395-f003:**
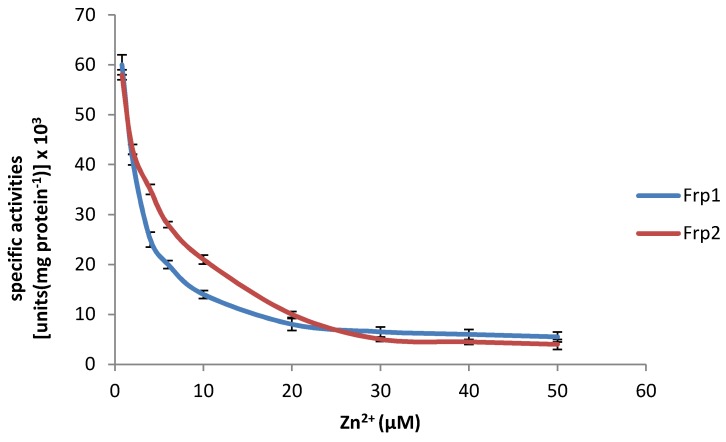
Activity of highly purified preparations of Frp1 and Frp2 in the presence of increasing levels of Zn^2+^.

**Figure 4 microorganisms-07-00395-f004:**
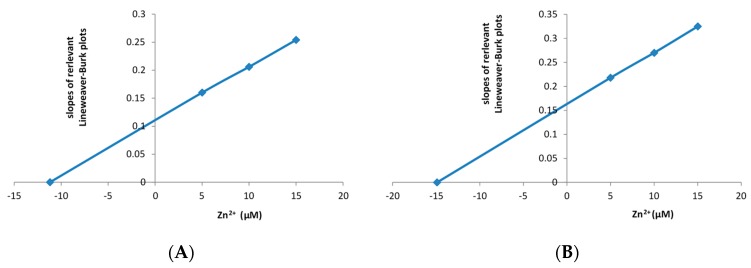
Re-plots of the slopes of the Lineweaver–Burk plots (Appendix A) for highly purified (**A**) Frp1 and (**B**) Frp2 at varying concentrations of Zn^2+^.

**Figure 5 microorganisms-07-00395-f005:**
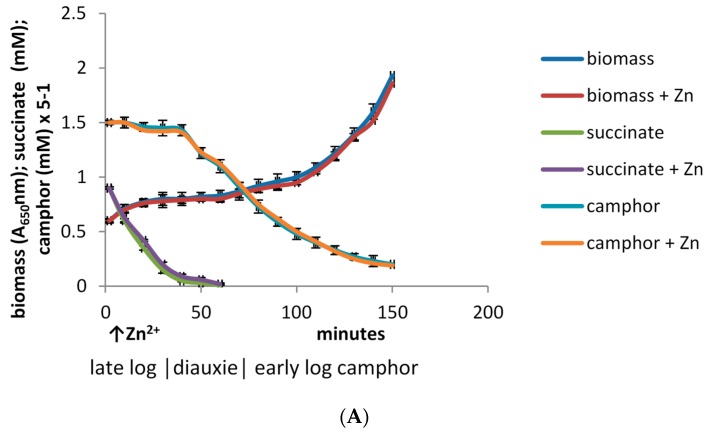
(**A**).Changes in the biomass (A_650_nm) of *P.putida* ATCC 17453 following addition of 30 μM Zn^2+^ in the diauxic interlude during growth on succinate-(*rac*)-camphor minimal medium. (**B**) Changes in the activity of the enantiocomplementary DKCMOs of *P.putida* ATCC 17453 following addition of 30 μM Zn^2+^ in the diauxic interlude during growth on succinate-(*rac*)-camphor minimal medium. (**C**).Changes in the activity of the cytP450MO of *P.putida* ATCC 17453 following addition of 30 μM Zn^2+^ in the diauxic interlude during growth on succinate-(*rac*)-camphor minimal medium. (**D**). Changes in the combined Frp1 + Frp2 activity of *P.putida* ATCC 17453 following addition of 30 μM Zn^2+^ in the diauxic interlude during growth on succinate-(*rac*)-camphor minimal medium. In each case, 0–40 min corresponds to late log phase growth on succinate, 41–70 min corresponds to the diauxic interlude, and 71–150 min correspond to early log phase growth on (*rac*)-camphor.

**Table 1 microorganisms-07-00395-t001:** Naturally occurring degradative plasmids in pseudomonads.

Plasmid	Degradative Pathway	Conjugative or Non-Conjugative	Approx. Size (Daltons × 10^6^)	Key Reference
CAM	Camphor	Conjugative	300	Rheinwald et al. [1]
OCT	*n*-Octane	Non-conjugative	250	Shapiro et al. [2]
SAL	Salicylate	Conjugative	42–55 (various)	Chakrabarty [3]
NAH	Naphthalene	Conjugative	46	Yen and Gunsalus [4]
TOL	Toluene/xylene	Conjugative	76	Duggleby et al. [5]
XYL-K	Xylene/toluene	Conjugative	90	Friello et al. [6]
2-HP	2-Hydroxypyridine	N.D.	63	Weinberger and Kolenbrander [7]
NIC	Nicotine/nicotinate	Conjugative	N.D.	Thacker and Gunsalus [8]
pOAD2	6-Aminohexanoic acid cyclic dimer	Non-conjugative	29	Fisher et al. [9]
pJP1	2,4-dichlorophenoxy acetic acid	Conjugative	58	Don and Pemberton [10]
pAC8	Xylene/toluene	Conjugative	76	Chakrabarty et al. [11]
pAC21	*p*-Chlorobiphenyl	Conjugative	65	Chatterjee and Chakrabarty [12]
pAC25	3-Chlorobenzoate	Conjugative	68	Chatterjee and Chakrabarty [12]
pAC27	3- and 4-Chloro-benzoate	Conjugative	59	Chatterjee and Chakrabarty [12]

N.D., not determined.

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
