# Peer review of "Conferring the Metabolic Self-Sufficiency of the CAM Plasmid of Pseudomonas putida ATCC 17453: The Key Role of Putidaredoxin Reductase"

_microorganisms, 2019, doi:10.3390/microorganisms7100395_

Round 1

Reviewer 1 Report

The author describes evidence that putative FMN reductase encoded on the CAM plasmid of Pseudomonas putida are sufficient to serve as redox partner for the monooxygenase contributing to camphor metabolization. Hence, this study shows for the first time, that the complete catabolic pathway is encoded on the megaplasmid without the need for chromosomal factors.

The study is well written and the strategy to assess contributions of chromosome and plasmid encoded reductase by specific chemical inhibition to the catabolic pathway is clearly outlined.

However, there are some major point which I think should be adressed:

Figures 2 to 5: I think,  specific measures of dispersion should be shown in the figures for each experiment. What does "high degree of consistency" as stated in the method section, l.139, mean? Already established methods should also be described briefly  in accord with the instructions for authors. This applies in particular for section in lines 126-136. Numbering of the method section is inconsistent. 2.3 and 2.4. appear two times (2.3: l.120 and l.129; 2.4.: l.126 and l.133) References 23 and 32: Why is this data not shown as supplement, if it supports the conclusion drawn. Can you comment or show experimental data, if this camphor metabolism can be implemented in other Pseudomonas/Proteobacteria if the plasmid encoded genes are sufficient for that as shown by the author and the plasmid is conjugative?

Minor:

Table1: Inclusion of specific references for the plasmids in the interesting overview on degradative plasmids would be useful.

l.259 Pseudobactin produced in the experimental setup of the study is also a nonribosomal peptide, isn't it?  And many of the P, putida strain NRP are functional analogues of pseudobactin and serve as siderophore. Hence thy may also be produced already in early growth stages. It may strengthen the point discussed here, if the assumend "target NRPs" would be named here.

l. 352. Typo in author names, I think.

Author Response

I would like to thank Reviewer 1 for their helpful comments and suggestions which contribute to significantly improving the quality of microorganisms-587899.

I will address each of the points raised as follows:

Reproducibility/standard error bars.

           This matter is addressed by the following change to Section 2.7 of the in the Materials and Methods, and the inclusion of standard deviation error bars in Figures 3 and 5A-D incl. Error bars are not included for Figures 4A and 4B because the relevant experimental protocols were only conducted once due to the unequivocal striking effect of Zn2+ on Frp1 and Frp2 as shown in Figure 3. Error bars are not included if Figure 2 because the Figure is taken from an earlier publication now cited in the relevant text box.

Extra details needed in the Materials and Methods section.

             Extensive relevant additional information has been included in address this issue and complement the citations to previous fully described methods and procedures. The numbering of the Sub -sections has also been corrected to introduce consistency.

Data referred to as References 23 and 32.

            The relevant data has now been included as Figure S2 and Figure S3 respectively in the expended Supplementary Materials section.

Conjugative nature of the CAM plasmid.

         The transmissible nature of the CAM plasmid both to other non-camphor catabolising strains of P. putida and other species of fluorescent pseudomonads was the basis on which Jim Rheinwald originally demonstrated the conjugative nature of this large extrachromosomal genetic element, and the relevant experiments and outcomes are fully reported both in his MSc thesis (Reference 13 in text) and subsequently published papers widely available in the public domain (References 1 and 15 in text).

Inclusion of relevant references in Table 1.

              This issue has been addressed by inclusion of new references 1 -12 in Table 1 and the bibliography section.

The nature of pseudobactin and its role in P. putida ATCC 17453.

         Pseudobactin is a NRP generated by P.putida by an as yet uncharacterised control mechanism triggered by low Fe(III) availability - it can be produced by relevant trophophasic and idiophasic culture conditions. Other secondary metabolites of P.putida more characteristic of idiophasic culture conditions are various polyketides, arylpolyenes, acyl-homoserine lactones and rhamnolipids, and reference to these classes of idiosynchratic metabolites has now been included in text.

Minor typos throughout the text.

         L352 ‘Thies’ and other typos in the original submitted text have now all been corrected and eliminated from the revised text (I hope!).

Text/etc added to the manuscript indicated in red, text/etc withdrawn from the manuscript indicated by strike through

Reviewer 2 Report

The main objective of this manuscript is to investigate the ability of the CAM plasmid as an autonomous extrachromosomal genetic element able to ensure entry of the C10 bicyclic terpene into the central pathways of metabolism via isobutyryl-CoA. The manuscript is well laid out and the content merits publication. The manuscript is clearly written and arranged. It presented new data about the metabolic potential of the CAM plasmid. The information provided could be of interest for the scientific community and the readers of “Microorganisms”. However, I have several points which should be taken into account prior to publication:

In the Materials and Methods section, it was written that “…all results are presented as averaged values…..”, however in each Figure there is a lack of standard deviation values. Please add them. In the Results section, the author should add one Figure presenting the growth of the cultivated bacteria under the applied conditions. Many times in the manuscript the author mentioned about the growth phases of bacteria, it is essential to present these data in Figure/Figures. Figure 1 – please write “O2”, “H2O” using subscripts (O2, H2O). Furthermore, this figure is not presented in a good quality and seems to be incomplete (at the bottom of the figure “3 x ace…..” was cut) Figure 2 – the reference should be added. Page 4 line 97 – change a bracket into [10] Page 4 line 112 – please add the name of the company and country Page 6 line155 – change a bracket into [12] Page 6 line 170 and 172 – please explain the abbreviations: Vmax, Km, Ki , they were mentioned for the first time in the text. Page 6 line 173 – please write Figure 4A and 4B Page 7 line 180 – delete (A, B) Page 8 line 200, 202 – „MW” should be written using subscripts as "MW" Page 8 line 204 – “Zn2+” should be written in superscript as "Zn2+" Page 7 line 214 – please add “+” to “Zn2 - supplemented” Page 8 line 237 – a lack of Figure 4C and 4D, these figures have been cited in the text. Page 9 line 259, 261 – change a bracket into [30,31] and [32]

To sum up, I recommend publication of the manuscript after major revision.

Author Response

I would like to thank Reviewer 2 for their helpful comments and suggestions which contribute to significantly improving the quality of microorganisms-587899.

I will address each of the points raised as follows:

Reproducibility/standard error bars.

           This matter is addressed by the following change to Section 2.7 of the in the Materials and Methods, and the inclusion of standard deviation error bars in Figures 3 and 5A-D incl. Error bars are not included for Figures 4A and 4B because the relevant experimental protocols were only conducted once due to the unequivocal striking effect of Zn2+ on Frp1 and Frp2 as shown in Figure 3. Error bars are not included if Figure 2 because the Figure is taken from an earlier publication now cited in the relevant text box.

Extra details needed to explain the growth phases of P.putida ATCC 17453 on succinate-(rac)-camphor minimal medium and their significance to the data presented.

          This issue is most relevant to the data presented in Figures 5 A – D and Supplementary Materials Figure 3. All of these Figures have now been amended to address this issue

Figure 1.

            The Figure has been reformatted to include the amendments requested4. Conjugative nature of the CAM plasmid.

Figure 2.

           The relevant reference has been cited in the Legend box.

Page 4, l 97; page 4, l 112; page 6 l 155; page 6, line 170 and line 172; page 6 line 173; page 7 line 180; page 8 line 200 and line 202; page 8 line 204; page 8 line 214; page 8 line 237; page 9 line259 and line 261.

       All of the above requested typographical changes have been made.

Text/etc added to the manuscript indicated in red, text/etc withdrawn from the manuscript indicated by strike through

Round 2

Reviewer 1 Report

The author addressed the concerns properly. The study  now represents another valuable brick towards the genetic and mechanistic understanding of camphor degradation by P. putida.

Reviewer 2 Report

The manuscript has been significantly improved and now warrants publication in "Microorganisms".